



# A global daily gap-filled chlorophyll-a dataset in open oceans during 2001–2021 from multisource information using convolutional neural networks

Zhongkun Hong[1,2], Di Long[1,2*], Xingdong Li[1,2], Yiming Wang[1,2], Jianmin Zhang[1,2], Mohamed A.Hamouda[3,4] and Mohamed M.Mohamed[3,4]

1. State Key Laboratory of Hydroscience and Engineering, Department of Hydraulic Engineering, Tsinghua University, Beijing, China
2. Department of Hydraulic Engineering, Institute of Ocean Engineering, Tsinghua University, Beijing 100084, China
3. Department of Civil and Environmental Engineering, United Arab Emirates University, Al Ain, United Arab Emirates
4. National Water and Energy Center, United Arab Emirates University, Al Ain, United Arab Emirates

(*Correspondence to Di Long at dlong@tsinghua.edu.cn)

**Abstract.** Ocean color data are essential for developing our understanding of biological and ecological phenomena and processes, and also important sources of input for physical and biogeochemical ocean models. Chlorophyll-a (Chl-a) is a critical variable of ocean color in the marine environment. Quantitative retrieval from satellite remote sensing is a main way to obtain large-scale oceanic Chl-a. However, data missing is a major limitation in satellite remote sensing-based Chl-a products, due mostly to the influence of cloud, sun glint contamination, and high satellite viewing angles. The common methods to reconstruct (gap filling) missing data often consider spatiotemporal information of initial images alone, such as data interpolation empirical orthogonal function, optimal interpolation, Kriging interpolation, and extended Kalman filter. However, these methods do not perform well in the presence of large-scale missing values in the image and ignore the potential of other information on missing pixels in the data reconstruction. Here we developed a convolutional neural network (CNN) named OCNET for Chl-a concentration data reconstruction in open ocean areas, considering environmental variables that are associated with ocean phytoplankton growth and distribution. Sea surface temperature (SST), salinity (SAL), photosynthetically active radiation (PAR), and sea surface pressure (SSP) from reanalysis data and satellite observations were selected as the input of OCNET to correlate with the environment and phytoplankton mass. The developed OCNET model achieves good performance in the reconstruction of global ocean Chl-a concentration data, and captures temporal variations of these features. This study also shows the potential of machine learning in large-scale ocean color data reconstruction and offers the possibility to predict Chl-a concentration trends under a changing environment.

**Key words.** Chlorophyll-a; U-Net; Satellite remote sensing; Data reconstruction



## Introduction

Chlorophyll-a (Chl-a), the primary pigment responsible for photosynthesis in plants, plays a vital role in the global carbon cycle and serves as a key indicator of the health and productivity of aquatic ecosystems (Righetti et al., 2019; Sun et al., 2021; Mouw et al., 2016). Chl-a is a measure of the amount of phytoplankton present in water bodies, and changes in its concentration can indicate shifts in the balance of these ecosystems, including the onset of harmful algal blooms or declines in productivity (Ho et al., 2019). Accurate and timely measurement of chlorophyll-a concentrations is therefore of great importance for

understanding and predicting the carbon fluxes and other elemental cycles in the oceans (Salgado-Hernanz et al., 2019; Laufkotter et al., 2016).

In recent years, satellite remote sensing has become a widely used method for monitoring chlorophyll-a concentrations on a global scale (Hu et al., 2012; Hu et al., 2019a; Feng et al., 2021). Satellite sensors can provide synoptic coverage of large areas, with a temporal resolution that ranges from daily to monthly. However, there are a lot of missing data in satellite products

caused by cloud, sun glint contamination, and high satellite viewing angles (Feng and Hu, 2016; Mikelsons and Wang, 2019). For example, there are over 70% missing data in global daily ocean color products from MODIS-Terra/Aqua and VIIRS-SNPP (Fig.1) (Feng and Hu, 2016; Liu and Wang, 2018). In addition, the spatial and temporal resolution of these measurements is often limited, and they are subject to various sources of error and uncertainty. These include atmospheric effects, such as scattering and absorption of light, which can distort the signal and introduce biases in the measurements (Hu et al., 2019a;

Zheng and Digiacomo, 2017). To address these limitations, it is useful to combine satellite remote sensing data with other sources of information, such as in situ measurements, model output, and ancillary data (Nikolaidis et al., 2014). Conventional methods for reconstructing missing data, such as data interpolation, DINEOF (Data Interpolating Empirical Orthogonal Functions), optimal interpolation, Kriging interpolation, and extended Kalman filter, often rely on the spatiotemporal information of the initial images alone (Wang and Liu, 2014; Hilborn and Costa, 2018; Catipovic et al., 2023; Liu and Wang,

2018). However, these geostatistical methods are not always effective in the presence of large-scale missing values and do not take into account the potential contribution of other information to the reconstruction of missing pixels (Konik et al., 2019).

The development of robust and efficient methods for synthesizing and integrating multisource information is becoming increasingly important as the availability and diversity of data sources continue to grow (Li et al., 2020). The integration of multisource information is not a trivial task, as the data sources may have different spatial and temporal scales, resolutions,

and uncertainties, and may be subject to different biases and errors. These differences can make it challenging to reconcile and combine the data in a meaningful and reliable way (Catipovic et al., 2023). With the proliferation of sensors and platforms, the volume of data being generated is increasing at an exponential rate, making it difficult to manage and analyze in a traditional way. Machine learning techniques, such as convolutional neural networks (CNNs), offer a promising approach for handling and extracting meaningful insights from this large and complex data stream (Zhang et al., 2018). CNNs are a class of deep

learning algorithms that have proven to be highly effective for image recognition and analysis tasks. They are particularly well suited to this problem, as they can automatically learn features and patterns from data and can handle large amounts of data



with high dimensionality and complexity. CNNs have been applied to a wide range of remote sensing applications, including the analysis of satellite imagery and the integration of multisource data. A number of studies have demonstrated the effectiveness of CNNs for analyzing global or regional daily chlorophyll-a products (Cao et al., 2020; Jin et al., 2021; Cen et al., 2022; Yussof et al., 2021). Here we propose a CNNs-based approach named OCNET for the reconstruction of global daily chlorophyll-a products from multisource information.

The OCNET model developed here is an improved version based on the general U-Net. One advantage of U-Net is its ability to handle large images while maintaining high-resolution segmentation results (Li et al., 2020; Ronneberger et al., 2015; Andersson et al., 2021). This is achieved by using skip connections, which allow the network to "skip" certain layers and merge higher-resolution information from early layers into the final prediction (Ronneberger et al., 2015; Wagle et al., 2020). This helps preserve fine-grained details of the input image and generates more accurate segmentation results (Krug et al., 2017). Here we utilized this characteristic of OCNET for global-scale input of big data, and successfully accomplished the task of data reconstruction. Given that the input image contains multi-level information elements at the global scale, it places high demands on how the model extracts feature information and captures its inherent correlations (Moran et al., 2022; Chen et al., 2019). Another advantage of U-Net is its ability to utilize contextual information from the entire image. Compared to other machine learning methods such as multiple linear regression and random forest, U-Net excels in learning complex nonlinear relationships between input data and output predictions (Ronneberger et al., 2015; Li et al., 2020). This is due to the use of nonlinear activation functions and the ability to learn hierarchical features through convolutional layers. Because artificial neural networks (ANNs) often face limitations in processing large images and struggle to incorporate global backgrounds into their predictions (Catipovic et al., 2023), U-Net outperforms traditional ANNs in various image segmentation tasks. Unlike ANNs, U-Net can handle high-resolution images and effectively incorporate global context information into its predictions (Andersson et al., 2021; Li et al., 2020).

In the big-data era, the effective integration and utilization of multisource information on the ocean are of importance for studying ocean color. The primary objective of this study was to propose the OCNET model which could be trained with environmental variables that are associated with ocean phytoplankton growth and distribution, in order to reconstruct high-quality gap-filled Chl-a data in open oceans. The Chl-a dataset covers the period from 2001 to 2021, with a daily temporal resolution and a spatial resolution of 0.25 °. Compared to traditional interpolation methods, this approach takes full advantage of environmental information mainly provided by ERA5 data, and considers the key factors that influence the growth and distribution of surface phytoplankton in the oceans. Furthermore, this method is not limited by the size of the ocean region or the temporal span covered by satellite data. By providing reliable environmental information, OCNET enables the retrospective analysis of Chl-a concentration data from the pre-satellite era and the prediction of future changes in global marine phytoplankton.



## 2. Data and methodology

### 2.1 Training data considerations

The Ocean-Colour Climate Change Initiative (OCCCI) version 5 and National Oceanic and Atmospheric Administration multi-sensor DINEOF global gap-filled data (termed as NOAA MSL12 hereafter) are two Chl-a products used in training the OCNET model (Table 1). OCCCI's data sources include the Moderate Spectral Resolution Imaging Spectroradiometer (MERIS) sensor from the European Space Agency, the SeaWiFS (Ocean Observation Wide Field Sensor) and MODIS-Aqua (Moderate Resolution Imaging Spectroradiometer-Aqua) sensors from NASA, and the National Oceanic and Atmospheric

Administration's VIIRS sensor (Visible and Infrared Imaging Radiometer Suite) (Sathyendranath et al., 2019). Data can be obtained starting from 1997. The Multi-Sensor Level-1 to Level-2 (MSL12) is the NOAA official enterprise VIIRS ocean color data processing system (Liu and Wang, 2022). The NOAA MSL12 dataset provides near-real-time, gap-free global maps of chlorophyll-a concentration by merging data from VIIRS and OLCI-Sentinel-3A satellites and utilizing the DINEOF method to fill in missing pixels caused by clouds, sun glint, and other factors (Liu and Wang, 2022). The strength of this dataset lies

in its broader spatial coverage, showcasing more marine features in coastal and inland waters and enhancing data accuracy. In addition, Chl-a data from OLCI-Sentinel-3B have not been applied in the production of OCCCI V5 or NOAA MSL12 datasets. Therefore, Sentinel-3B data were used for the evaluation and comparison of the final performance of the OCNET model as an independent product.

**Table 1 Full names, spatiotemporal resolution, temporal coverage, sources, and other information of data used in this study.**

| Data | Variables | Abbreviation | Unit | Temporal resolution | Spatial resolution | Temporal coverage | References |
|---|---|---|---|---|---|---|---|
| OCCCI V5 | Chlorophyll a | Chl-a | mg/m$^3$ | daily | 4km | 1997.9.4–2021 | (Sathyendranath et al., 2019) |
| MODIS-Aqua | Photosynthetically Available Radiation | PAR | einstein/(m$^2$ d) | daily | 4km | 2002.7.4–present | https://oceancolor.gsfc.nasa.gov/l3 |
| MODIS-Terra | Photosynthetically Available Radiation | PAR | einstein/(m$^2$ d) | daily | 4km | 2000.2.24–present | https://oceancolor.gsfc.nasa.gov/l3 |
| VIIRS-SNPP | Photosynthetically Available Radiation | PAR | einstein/(m$^2$ d) | daily | 4km | 2012.1.2–present | https://oceancolor.gsfc.nasa.gov/l3 |
| OLCI S3B NRT | Chlorophyll a | Chl-a | mg/m$^3$ | daily | 4km | 2018.5.14–present | www.star.nesdis.noaa.gov |
| NOAA MSL12 | Chlorophyll a | Chl-a | mg/m$^3$ | daily | 9km | 2018.2.9–present | (Liu and Wang, 2022) |
| ERA5 | Surface Pressure | SSP | Pa | hourly | 0.25 ° | 1940.1.1–present | (Hersbach et al., 2020) |



| ERA5 | Sea Surface Temperature | SST | K | hourly | 0.25 ° | 1940.1.1–present | (Hersbach et al., 2020) |
|---|---|---|---|---|---|---|---|
| ORAS5 | Salinity | SAL | PSU | monthly | 0.25 ° | 1958.1.1–present | (Zuo et al., 2019) |
| ETOPO1 | Depth | Dep | m | – | 1' | – | (Information and Doc/Noaa/Nesdis/Ncei National Centers for Environmental Information, 2009) |
| WOA2013 | Salinity | SAL | PSU | – | 0.25 ° | – | (Zweng et al., 2013) |

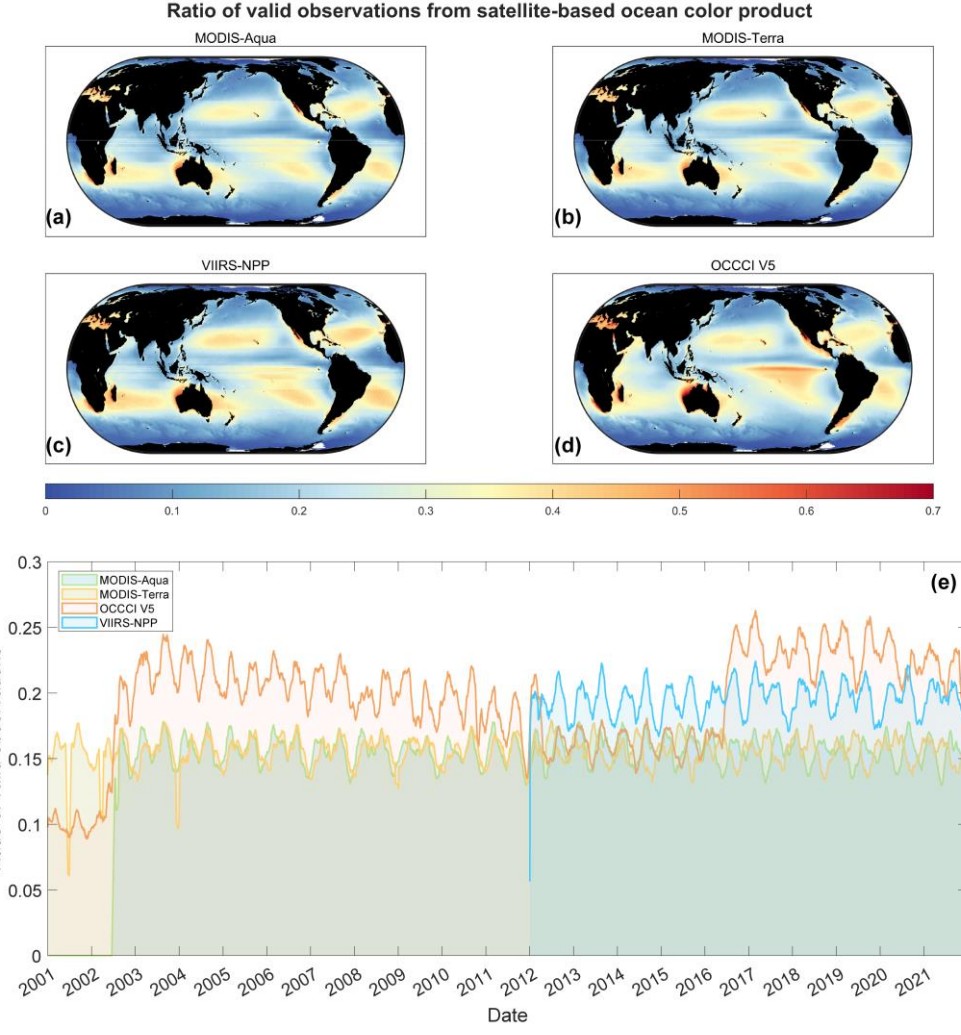

**Figure 1 Valid data proportion of each satellite-based Chl-a product during 2001–2021. The global distribution of valid chlorophyll-a (Chl-a) observations ratio was examined using (a) MODIS-Aqua, (b) MODIS-Terra, (c) VIIRS-SNPP, and (d) OCCCI satellite datasets, along with an examination of the (e) temporal variation over their respective coverage periods.**





The ocean Chl-a data of the OCCCI product cover more than 20 years. Compared with a single satellite product, OCCCI

products that integrate multiple sources of data improve data availability by complementing different data sources (Fig.1). Due to changes in satellite data sources used in different years, the valid data proportion of OCCCI varies greatly in different time periods. In addition, OCCCI has been significantly improved with the introduction of more satellite data. However, valid observations from OCCCI are unevenly distributed globally (Fig.1d). And missing data on more than 70% of satellite-based products still pose a huge obstacle to the study of ocean color (Feng and Hu, 2016). The NOAA MSL12 achieved the

spatiotemporal continuity of chlorophyll concentration products by the DINEOF method, but NOAA MSL12 are only available after February 9, 2018. Given the high coincidence of OCCCI and NOAA MSL12 datasets in the selection of satellite sources, these two datasets were selected as the main data sources. Other Chl-a data products from single-mission satellites, such as MODIS-Aqua/Terra and VIIRS-SNPP, which have more severe missing values (Fig.1), were only used for comparison in this study and were not directly applied.

We have selected three environmental variables, i.e., sea surface temperature (SST), salinity (SAL), and photosynthetically active radiation (PAR) as the input data for the OCNET model. These variables play a significant role in influencing the growth of marine phytoplankton (Flynn, 2001). SST and SAL are two important environmental factors influencing marine phytoplankton growth (Han and Zhou, 2022). SST affects algal metabolic rates, enzymatic activity, cell division rates, and growth cycles, among other biological processes (Nelson et al., 2020). Variations in salinity can influence osmoregulation in

marine phytoplankton and ion balance within cells (Nelson et al., 2020). Furthermore, from a hydrodynamic perspective, changes in wind patterns and ocean currents can also affect the distribution of surface algae. Therefore, we selected reanalysis data ERA5's SSP, SST, and Ocean Reanalysis System 5's SAL as input data for the OCNET model.

In addition to SST and SAL, PAR is a crucial energy source for plant photosynthesis, and its distribution is of great importance for studying plant growth and photosynthetic processes (Xing and Boss, 2021). Its spatiotemporal variations can impact the

photosynthetic efficiency, biomass accumulation, and yield of plants (Righetti et al., 2019). Here we selected PAR data from satellite sources, specifically MODIS-Terra/Aqua and VIIRS-SNPP, as part of the model input. To address spatial gaps in satellite data and correct biases among different datasets, preprocessing and fusion techniques were applied to the PAR data from different satellite products (see Section 2.2).

Both ETOPO1 and WOA13 data were used as auxiliary data for determining the study area and were not input for the OCNET

model. The ETOPO Global Relief Model is a global digital elevation model developed by the National Geophysical Data Center (NGDC), a NOAA department (Information and Doc/Noaa/Nesdis/Ncei National Centers for Environmental Information, 2009). It provides elevation data for the Earth's surface and finds applications in areas such as topographic maps, hydrological models, oceanography, and other related fields. Data of ETOPO1 were selected because of the 1-min resolution it offers. ETOPO1 is widely utilized in scientific and research communities due to its high accuracy, serving various purposes

like mapping, visualization, resource management, and environmental modeling (Moran et al., 2022; Righetti et al., 2019). The World Ocean Atlas 2013 (WOA2013) is a comprehensive collection of objectively analyzed climatology data for various oceanic parameters, including temperature, salinity, oxygen, phosphate, silicate, and nitrate (Zweng et al., 2013). It was



provided by NOAA's National Oceanographic Data Center - Ocean Climate Laboratory. Salinity data provided by WOA13 are often used as a reference to analyze abnormal variations in ocean salinity (Righetti et al., 2019; Li et al., 2017).

The study area considered here mainly focuses on the middle and low latitudes of the open ocean area, constrained primarily due to limitations in satellite data sources. In particular, satellite-based Chl-a products exhibit a substantial number of missing values in high latitudes and coastal regions (Fig.1). Additionally, the accuracy of chlorophyll concentration retrievals is affected mostly by the presence of high concentrations of suspended matter resulting from sediment discharge from rivers in coastal areas. To mitigate the influences stemming from complex coastal environments on the analysis of ocean color, we

excluded regions from seas shallower than 200 m and from seas with surface salinities below 25, as determined by ETOPO1 and WOA2013 datasets, respectively (Righetti et al., 2019).

## 2.2 Data preprocessing

For the OCCCI V5 data, we selected its climatology product as the background field. Because the OCCCI climatology data only provide valid observations for 12 months, temporal smoothing interpolation was performed to cover each ocean grid cell

from January 1, 2001, to December 31, 2021. Due to the presence of missing values in both the daily and monthly data products of OCCCI V5, it is not suitable for direct use as model input. Therefore, the climatology product without missing values in the spatial domain was used to set the Chl-a baseline.

As PAR data from different satellite sources were used in this study, preprocessing and bias correction were applied. The overlapping period of MODIS and VIIRS data from 2012 to 2021 was chosen as the reference, using a ratio-based method

with MODIS-Aqua as the baseline for bias correction. In cases of missing values in the spatial domain, the three different products were used for complementarity. If effective observational values were not available, linear spatial interpolation was performed. Finally, a spatiotemporally continuous PAR dataset was obtained for model input.

For the reanalysis datasets, as they are already spatiotemporally continuous with a spatial resolution of 0.25 °, no additional preprocessing is required. The average of the first five levels of SAL data (approximately 5.14 m) from ORAS5 was taken as

the input. It should be noted that ORAS5 has a spatial resolution of 9 km near the polar regions. However, this study does not consider the inversion of Chl-a data in high-latitude areas. Considering the different spatial resolutions of the data, apart from the reanalysis data, the other input data for the model in this study were resampled to 0.25 ° using the nearest interpolation method.

When using the data mentioned above as inputs for the OCNET model, normalization is necessary. For environmental variables

(SST, SSP, SAL, and PAR), normalization was performed according to Eq.1 where the parameters used in the formula were pre-calculated (Table 2). Due to the presence of numerous low values in the Chl-a concentration data in open waters, it is first natural logarithm transformed and then normalized to achieve a uniform distribution of the input data (Eq.2).

**Table 2 Maximum, minimum, and mean values obtained for the environmental variables.**

| Variables | max | min | mean | units |
|-----------|-----|-----|------|-------|



| SST | 310.06 | 269.17 | 286.821 | K |
|------|--------|--------|---------|------------------|
| SSP | 106980 | 54834 | 96643 | Pa |
| PAR | 70.329 | 0 | 32.2007 | einstein/(m² d) |
| SAL | 43.467 | 0 | 34.169 | PSU |

$$X_N = \frac{X - \overline{X}}{X_{max} - X_{min}} \qquad (1)$$

$$C_N = \frac{\ln(C) + 4.61}{4.61 \times 2} \qquad (2)$$

where $X$ represents different environmental variables, subscript N represents the normalized variables, subscripts "max" and "min" correspond to the maximum and minimum values in Table 2, and $\overline{X}$ represents the mean. C represents Chl-a data. Values of Chl-a concentration lower than 0.01 mg/m³ were all set to 0.01 mg/m³. Actually, the accuracy of satellite retrievals cannot reach such a small value.

**2.3 Model architecture**

Data-driven deep learning algorithms can extract high-level information from multisource input data using multiple non-linear processing layers (Li et al., 2020; Cen et al., 2022). In the research of large-scale, long-term, and multi-data scenarios, deep learning algorithms excel at discovering data patterns and inherent connections (Li et al., 2020; Andersson et al., 2021). Given the applicability of CNNs to satellite remote sensing imagery and climate model data, we constructed the global OCNET model consisting of 405 regional CNNs. Specifically, each CNN employed in the individual regions was based on the U-Net model

(Fig.2). U-Net, initially designed for medical image segmentation, is a variant of the CNN (Ronneberger et al., 2015). Across various applications, U-Net has been consistently proven to be highly effective in terms of learning accuracy and pixel-wise mappings (Andersson et al., 2021; Urakubo et al., 2019; Wagner et al., 2019).

Here we applied the OCNET to reconstruct global Chl-a concentration data in open ocean areas, considering environmental variables that are associated with ocean phytoplankton growth and distribution. SST, SAL, and SSP from reanalysis data and

PAR from satellite observations were selected as the input of OCNET to correlate with the environment and phytoplankton mass. The whole area considered in this study covers latitude 45 °N to 45 °S, and longitude 180 °W to 180 °E. The open ocean is divided into 45 horizontal and 9 vertical zones, 405 in total. Each area has a size of 16 ° × 16 ° and a side length of 64 grid cells. There is an 8 ° overlap in the latitude direction between each pair of adjacent regions at the same latitude. Additionally, there is a 6.25 ° overlap in the longitude direction between each pair of adjacent regions at the same longitude. This is to reduce

the boundary effect caused by dividing regions for network training separately.





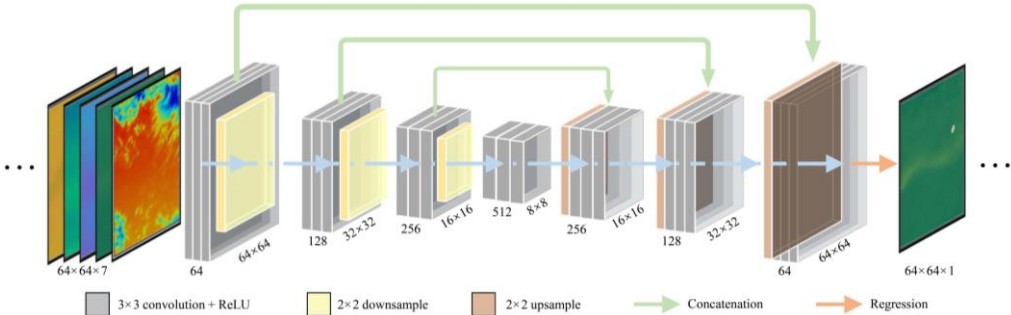

**Figure 2 Flowchart of the developed OCNET model in each zone. The OCNET model, comprised of deep learning U-Net models, receives three monthly averaged variables (SST, SAL, and PAR) and two daily real-time variables (SST and SSP) as input. The climatology Chl-a of OCCCI and daily Chl-a data of NOAA MSL12 were treated as background and target set, respectively.**

Inputs to the network include Chl-a_OCCCI, Chl-a_N, SST, SAL, SSP, PAR and SST_d. Chl-a_OCCCI is the climatology data from OCCCI, as the background field of the dataset with only one value per month. Considering the typical monthly growth cycle of phytoplankton, we calculated the environmental factors influencing marine algae growth by averaging the data from the preceding month as input variables. Therefore, SST, SAL, and PAR took the average of one month forward as input to OCNET. In addition, the values of SST_d and SSP were also taken as the input of the day, respectively.

There are totally 405 zones of size 64×64 globally. Each zone has its own independent U-Net. Each network undergoes a maximum of 100 training steps to ultimately output the network model for each region. First, the input data with a size of 64×64×7 are passed through the initial convolutional layer, which consists of 64 filters. Each filter has a grid size of 3×3 and a stride of 1. Subsequently, an activation function is applied to the data, and the dimension of the feature map is reduced to half of its original size, resulting in a size of 32×32×64, through a pooling layer operation of size 2×2. After completing this

initial step, the subsequent operations follow a similar pattern. The feature map undergoes a halving of its spatial dimension through pooling, while the number of channels is doubled through convolution. The final feature map obtained from these operations has a size of 8×8×512, and it serves as input for the subsequent decoding process. The decoding process mirrors the encoding process described earlier. It is important to note that the encoding and decoding networks are connected through skip connections, enabling the preservation of information that may be lost during downscaling. This U-Net structure facilitates

the preservation of detailed information from previous layers during the subsequent decoding stage. Finally, the last layer consists of a single filter that outputs a feature map with a size of 64×64×1, representing a single channel of data. Finally, by inputting environmental information from 2001 to 2021 into the OCNET model, a spatiotemporal continuous dataset of Chl-a concentration was reconstructed, covering the period from 2001 to 2021.

**2.4 Statistical tests**

**2.4.1 Evaluation of OCNET output**

In the simulation performed by OCNET, the data from the year 2021 was selected as the testing set. This portion of the data was excluded from model training and validation, and was solely used for evaluating the quality of the final data. The





commonly used evaluation metrics, including CC, bias, and RMSE, were employed for this purpose. The specific formulae

used for the calculations can be found in Table 3, while the evaluation results are presented in Section 3.2.

**Table 3 Statistical metrics used in evaluating the reconstructed Chl-a (C) against the observed data (Cg) from the NOAA MSL12 during the testing period. An overbar donates the mean during evaluation periods. N denotes the number of data pairs. Cov denotes the covariance and σ is the standard deviation.**

| Performance Score | Score symbol | Equation | |
|---|---|---|---|
| Pearson's correlation coefficient | CC | $CC = \dfrac{cov(C, C_g)}{\sigma(C)\sigma(C_g)}$ | (3) |
| Bias | Bias | $BIAS = \dfrac{\sum(C - C_g)}{\overline{C_g}}$ | (4) |
| Root mean square error | RMSE | $RMSE = \sqrt{\dfrac{\sum(C - C_g)^2}{N}}$ | (5) |

**2.4.2 Evaluation using the ETC method**

Due to the lack of enough and reliable in-situ measurements for the assessment of global ocean Chl-a, the extended triple

collocation (ETC) method was used to indirectly evaluate the quality of OCNET model output data (Mccoll et al., 2014). The

ETC method uses exactly the same assumptions as the triple collocation (TC) method. The TC method utilizes three mutually

independent datasets to assess the relative errors of the data without requiring the knowledge of the true value. This method

was initially developed by Stoffelen (1998) and has been widely used for soil moisture assessment (Dorigo et al., 2010; Miralles

et al., 2010). The ETC method, improved by Mccoll et al. (2014) from the TC method, provides the correlation coefficient as

another performance index. The ETC method has also been extensively applied, such as in the evaluation of sea surface

temperature data (Gentemann, 2014).

Because the Sentinel-3B data are not used in the OCCCI and NOAA MSL12 datasets, it was selected as an independent dataset

for evaluation. Chl-a data products from Sentinel-3B, NOAA MSL12 and OCNET were used in ETC method. Considering the

available time period of Sentinel-3B data, the evaluation covers the period from June 7, 2019, to December 31, 2021. Due to

the presence of numerous missing values in the Sentinel-3B data products, grid cells with severe missing values, i.e., grid cells

with fewer than 30 valid days, were excluded, and the remaining grid cells were retained for evaluation. It should be noted that

since OCNET was trained using NOAA MSL12 as the target set, they cannot be considered mutually independent datasets.

This evaluation mainly utilizes Sentinel-3B data as a third-party source to validate the reliability of the OCNET model. It is

possible that the results of the ETC in some grid cells may yield a negative square of the correlation coefficient or root mean

square error. This can happen if the sample size is too small, or if one of the assumptions of ETC is violated. In the final

presentation of results, these grid cells were excluded.

The calculation method is based on Eq.6-Eq.11, where $C_{ij}$ represents the covariance between the $i$-th and $j$-th data points. The

calculated correlation coefficient (tCC) and root mean square error (tRMSE) based on the TC method are denoted as $\rho$ and $\sigma$,





respectively. It should be noted that the magnitude of the tCC and tRMSE only reflects the relative performance as opposed to
the absolute values.

$$\sigma_{t,1} = \sqrt{C_{11} - C_{12}C_{13}/C_{23}} \tag{6}$$

$$\sigma_{t,2} = \sqrt{C_{22} - C_{21}C_{23}/C_{13}} \tag{7}$$

$$\sigma_{t,3} = \sqrt{C_{33} - C_{31}C_{32}/C_{21}} \tag{8}$$

$$\rho_{t,1} = \pm\sqrt{C_{12}C_{13}/C_{11}C_{23}} \tag{9}$$

$$\rho_{t,2} = \pm\mathrm{sign}(C_{13}C_{23})\sqrt{C_{12}C_{23}/C_{22}C_{13}} \tag{10}$$

$$\rho_{t,3} = \pm\mathrm{sign}(C_{12}C_{23})\sqrt{C_{13}C_{23}/C_{33}C_{12}} \tag{11}$$

## 3. Results

### 3.1 Spatial variations and trends in global Chl-a estimates during 2001–2021

We have developed high-quality gap-filled Chl-a data in open oceans using the OCNET model. The dataset covers the time
period from 2001 to 2021 and has a spatial resolution of 0.25, with a daily temporal resolution. We applied the natural logarithm
transformation to the Chl-a concentration values when generating maps (Fig.3). This transformation was necessary due to the
relatively low Chl-a concentrations in most sea areas but the relatively high concentrations in areas experiencing algal blooms.
It can be observed that there are high chlorophyll concentrations in the sea areas near the west coast of Africa (~2.2 mg/m$^3$),
the east coast of Asia (~1.1 mg/m$^3$), and the west coast of the Americas (~2.3 mg/m$^3$), which indicates a higher likelihood of
algal blooms in these regions. Chl-a concentrations near the equator and in regions above 30 $^\circ$latitude are higher than in open
ocean regions between 10 $^\circ$ and 20 $^\circ$latitude. In addition, oceanic regions far from the continents, such as the Pacific Ocean,
Indian Ocean, and Atlantic Ocean, exhibit low chlorophyll concentration distributions (less than 0.05 mg/m$^3$). This also
suggests a higher possibility of algal blooms in coastal areas to some extent.

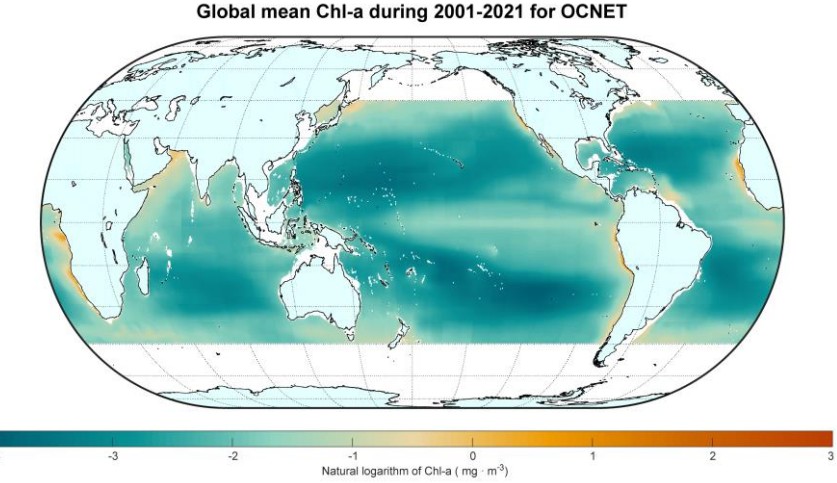

**Figure 3 Natural logarithm of the OCNET model output Chl-a during 2001–2021. Light blue represents land areas. White denotes areas that are not considered in this study.**

To ensure spatial continuity in the global Chl-a concentration product, the data underwent regional processing before being input into the OCNET model. Subsequently, overlapping region processing and image stitching were performed, resulting in a seamless global Chl-a concentration product without noticeable discontinuity or fragmentation. Although the OCNET model was trained separately for each region, the final results obtained after adequate data preprocessing and sufficient training steps were consistent and globally continuous. This outcome further highlights the effectiveness of the OCNET model in global data reconstruction.

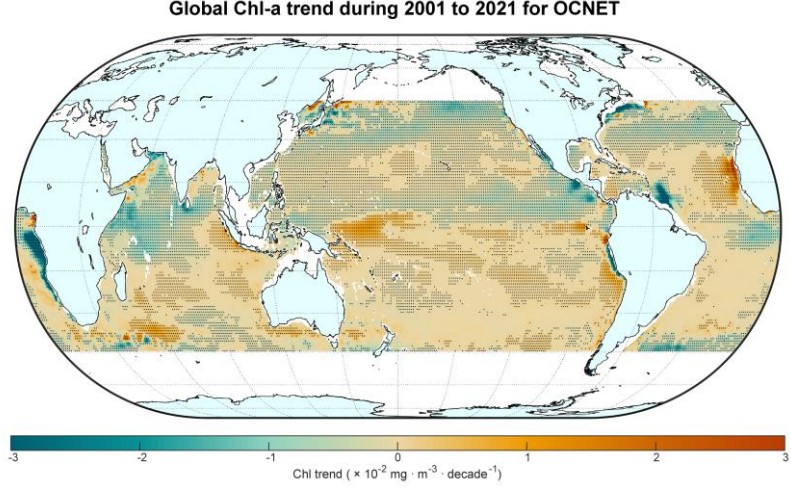

**Figure 4 Global Chl-a trends from OCNET over the period Jan 2001–Dec 2021. Regions with significant trends ($p<0.05$) are marked with black dots.**

Trends in Chl-a concentration in the global ocean area from 2001 to 2021 according to the output of the OCNET model were derived (Fig.4). To emphasize regions exhibiting clear trends, data in this section were not subjected to natural logarithm





transformation and was magnified instead (please note the unit is $10^{-2}$ mg m$^{-3}$ decade$^{-1}$). It can be observed that, in general, the sea areas closer to continental land exhibit more significant trends. Although the sea areas near the west coast of Africa show high chlorophyll concentrations (Fig.3), the two hemispheres, northern and southern, exhibit different trend patterns.

Specifically, the sea areas on the western side of the northern hemisphere of Africa show a clear upward trend in chlorophyll concentration ($\sim$4$\times$10$^{-2}$ mg m$^{-3}$ decade$^{-1}$), while the sea areas on the western side of the southern hemisphere show a significant downward trend ($\sim$-8$\times$10$^{-2}$ mg m$^{-3}$ decade$^{-1}$). The sea areas near North America predominantly exhibit a noticeable downward trend ($\sim$-5$\times$10$^{-2}$ mg m$^{-3}$ decade$^{-1}$). The islands around the northern part of South America show a pronounced decrease in chlorophyll concentration ($\sim$-5$\times$10$^{-2}$ mg m$^{-3}$ decade$^{-1}$), while the sea areas on the western side exhibit distinct increasing or

decreasing trends at different latitudes. The chlorophyll concentration variation around Japan in eastern Asia shows the most significant trend. The sea areas near Japan demonstrate a decrease in chlorophyll concentration at lower latitudes and an increase at higher latitudes. In general, there are more areas in the open oceans worldwide where Chl-a concentration shows a decreasing trend than areas where it shows an increasing trend.

**3.2 Temporal variations in Chl-a estimates in different ocean regions**

To facilitate the analysis and evaluation of regional data, we divided the study area into 10 regions based on latitude, longitude, and the ranges of oceans (Fig.5). The division of sea areas considered the characteristics of the regions and the influence of ocean currents, taking into account the division of biogeochemical provinces (Reygondeau et al., 2013). To avoid excessive complexity resulting from overly detailed regional divisions, a final selection of 10 regions was determined. This study calculated and presented the Chl-a concentration products for these 10 regions in a 20-year time series (Fig.5). Due to the

OCNET model's target dataset being NOAA MSL12, the output results of the OCNET model are consistent with NOAA MSL12 after February 9, 2018. However, it can be observed that the results of the OCNET model are noticeably lower than the results of OCCCI V5, particularly in regions 2, 4, 6, 7, 8, and 9. The primary reason for this systematic bias is the discrepancy between the NOAA MSL12 data and the OCCCI V5 data products.





Figure 5 Global open ocean was divided into 10 regions in this study, and the temporal variations of Chl-a from 2001 to 2021 are shown for each region. The blue line represents the output results of the OCNET model, the red line represents the results from OCCCI V5, the green line represents the results from NOAA MSL12 data, and the dark dashed line represents the linear fit of OCNET. The trends of OCCCI V5 and the OCNET model outputs during 2001 to 2021 are indicated with their respective color



**labels in the top left corner of the temporal variation plot. For comparison purposes, we only consider and display calculations based**
**on grid cells with valid values from OCCCI V5.**

It can be observed that the long-term Chl-a concentration trends in most regions are relatively small, with changes within 0.001

mg/m$^3$ per year, except for Region 3. In terms of seasonal variations, regions 3, 5, and 9 exhibit larger intra-annual fluctuations.

On the other hand, regions 4, 7, and 8, which encompass a wider range of low Chl-a concentrations (Fig.3), show smaller

seasonal fluctuations. It is worth noting that OCCCI V5 and OCNET show significant deviations in Region 9, where there are

higher Chl-a concentrations (particularly during the period from 2010 to 2015). Considering that Region 9 mainly covers the

sea areas surrounding the Americas (Fig.5), it is likely influenced by human activities. Additionally, the satellite retrieval of

Chl-a concentration data in this region is of poorer quality due to high sediment concentrations and turbidity near the coastline.

This partially explains the significant interannual variability observed in OCCCI V5 products for Region 9. Furthermore, both

OCCCI V5 and NOAA MSL12 products have instances of unusually high Chl-a values, such as in regions 7 and 10 for OCCCI

V5, and Region 4 for NOAA MSL12. These abnormally high Chl-a concentrations, which surpass typical values for the

respective years, could be due to algal blooms or satellite data quality issues. Overall, from the long-term trends, most regions

show small magnitudes of change, with more regions exhibiting a decreasing trend.

### 3.3 Evaluation of OCNET's performance

The target dataset for the OCNET model is the NOAA MSL12 data product with a time span from February 9, 2018, to

December 31, 2021. Details of model construction are explained in Section 2.3, where the data were divided into training,

validation, and testing sets in a ratio of 7:1.5:1.5. Three statistical metrics, i.e., CC, bias, and RMSE, were selected to evaluate

the training performance of the OCNET model (Fig.6 and Table 4) for different regions (Fig.5).



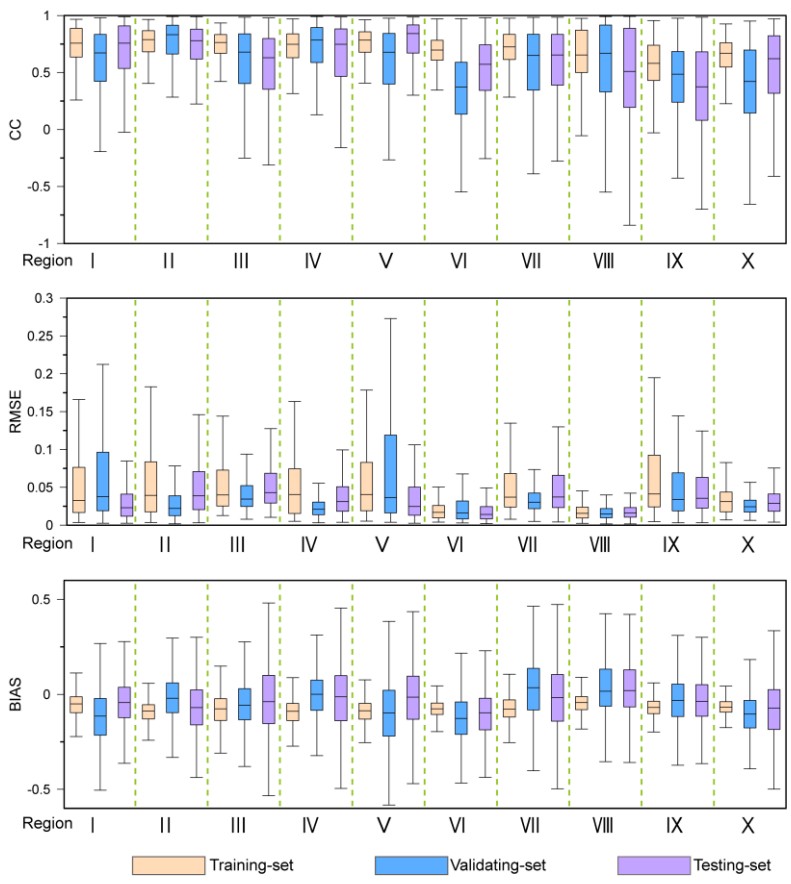

**Figure 6** Boxplot of evaluation results of the OCNET model in each region.

**Table 4 Chl-a concentration change rates for each region from three datasets and the median values of evaluation metrics for the OCNET model.**

| Region | Change rate ($\times 10^{-4}$ mg $\cdot$ m$^{-3}$ $\cdot$ year$^{-1}$) | | | Evaluation Index | | |
|---|---|---|---|---|---|---|
| | OCNET | OCCCI | NOAA MSL12 | CC | Bias | RMSE |
| 1 | -3.5 | -1.1 | -103.3 | 0.73 | -0.06 | 0.03 |
| 2 | -7.1 | -1.9 | -1.5 | 0.78 | -0.08 | 0.04 |
| 3 | -13.3 | -16.7 | -91.2 | 0.75 | -0.07 | 0.04 |
| 4 | -4.0 | 7.1 | -17.6 | 0.72 | -0.08 | 0.04 |
| 5 | -5.9 | -4.9 | -94.4 | 0.76 | -0.09 | 0.04 |
| 6 | -3.5 | -8.6 | 3.5 | 0.66 | -0.09 | 0.02 |
| 7 | -3.1 | 3.2 | -26.0 | 0.71 | -0.06 | 0.04 |
| 8 | -0.8 | 1.0 | 6.1 | 0.63 | -0.03 | 0.02 |
| 9 | -1.6 | -6.1 | -96.0 | 0.56 | -0.07 | 0.04 |



| 10 | -3.5 | -9.1 | -29.8 | 0.64 | -0.08 | 0.03 |
|---|---|---|---|---|---|---|

From the daily evaluation, it can be seen that the model performs well (Fig.6). The median values of CC for the training set are mostly above 0.6. The performance of the validation set and the test set is similar, but individual regions show poor performance. For example, in the validation set, the median values of CC for regions 6, 9, and 10 are around 0.4 and 0.5, and for region 9 in the test set, the median value of CC is around 0.4. This corroborates the findings in Section 3.1 that region 9, being mostly near the American continent, is heavily influenced by human activities, and the satellite data quality in coastal areas is also poorer. In terms of bias, the performance of the training set is excellent, with biases within a small range for each region. The boxplot ranges for the validation set and the test set also fluctuate within 0.2. It is worth noting that, due to the low Chl-a concentrations in most marine areas, the calculated biases are defined as relative biases (with the denominator being the mean of the target dataset). Therefore, it is possible to have higher biases in regions with low Chl-a concentrations. For the RMSE, both the training set, validation set, and test set are below 0.2, with most of them below 0.1, indicating excellent performance. Regions 6 and 8 have the lowest RMSE values. This may be because regions 6 and 8 mostly cover low Chl-a concentration offshore areas with minimal seasonal fluctuations (Fig.5).

According to the results of the Chl-a concentration rate of change in each region, it can be observed that most regions show relatively small trends (Table 4). Most regions exhibit a decreasing trend, which is consistent with the conclusions of existing related studies (Le Grix et al., 2021; Beaulieu et al., 2013; Signorini et al., 2015). Based on the results of the OCNET model, regions 2, 3, and 5 show larger decreasing magnitudes, while the other regions also exhibit a decreasing trend. According to the results of OCCCI, except for regions 4, 7, and 8, which show small increasing trends, the other regions demonstrate a decreasing trend, with regions 3, 6, 9, and 10 showing more pronounced declines. As for NOAA MSL12, except for regions 6 and 8, which show an upward trend, the other regions display a decreasing trend. Due to the relatively short time series of NOAA MSL12, it cannot reflect long-term trend characteristics. It can be seen that NOAA MSL12 shows a significant decrease in Chl-a concentration in regions 1, 3, 5, 7, 9, and 10. This overall decline exhibited by NOAA MSL12 directly influences the training results of the OCNET model. Therefore, OCNET and OCCCI share similarities in long-term trends but may have differences in individual regions.



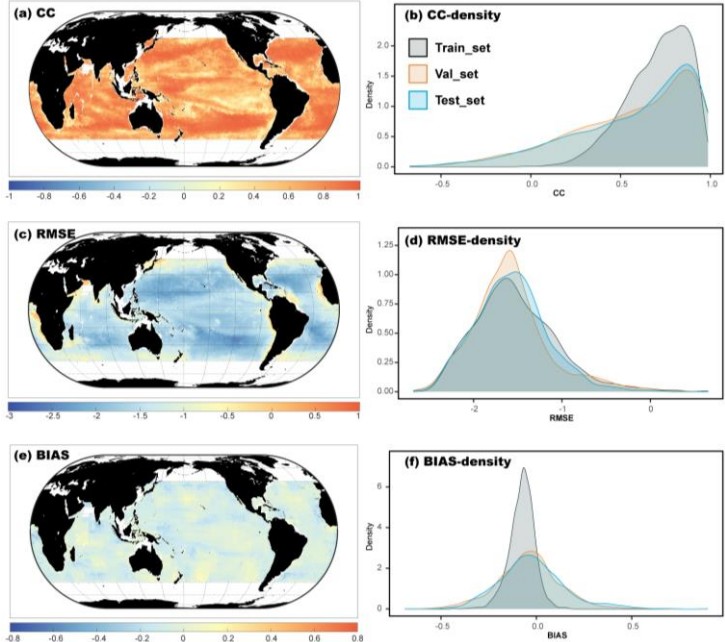


**Figure 7 Evaluation results of OCNET based on the NOAA MSL12 dataset. a), c), and e) represent the global distribution maps of CC, RMSE, and bias from February 9, 2018, to December 31, 2021, respectively. b), d), and f) display the density distributions of the evaluation results for CC, RMSE, and bias across the training, validation, and testing sets, respectively. Note that RMSE took logarithm base 10.**

From the comparison results with the target data NOAA MSL12 (Fig.7), the OCNET model has effectively learned the relationship between environmental data and Chl-a concentration variations. At the global scale, the overall performance of CC is good, with most regions above 0.7. Regions with lower CC are mainly concentrated in the central Pacific, where the OCNET model output shows apparent systematic biases compared to OCCCI (Fig.7a). Due to the lower mean Chl-a concentration in Region 8 (Fig.3), its RMSE and bias performance are also better (Fig.7 (c-e)). The preliminary evaluation

results in Region 8 suggest that OCNET's performance is not as good as in other areas. This may be related to the specific climate characteristics or low satellite data quality in that region. The complex factors ultimately result in OCNET's less optimal learning effect in Region 8. For the density distribution maps of the evaluation results for the training, validation, and testing sets, the performance of the training set is generally excellent. The performance of the validation and testing sets is comparable. From the results of bias, the training set shows a clear tendency of underestimation (Fig.7d), while the validation

and testing sets also exhibit some underestimation but less pronounced. This may be due to the smoothing effect of OCNET on some abnormally high values in the satellite data (Section 3.2). In summary, the evaluation results indicate that OCNET performs exceptionally well in the reconstruction of global Chl-a concentration data.

Earth System
Science
Data

## 3.4 Extended triple collocation evaluation

Output of the OCNET model, NOAA MSL12, and Sentinel-3B's Chl-a concentration data were selected for the ETC evaluation
method (Fig.8). It should be noted that the Sentinel-3B dataset was considered independent of the other two datasets, while
output of OCNET is not independent of the NOAA MSL12 dataset. Therefore, the evaluation results are biased towards
OCNET and NOAA MSL12 data and may underestimate Sentinel-3B data. The purpose of the TC method evaluation here
was to demonstrate the quality of OCNET output data compared to NOAA MSL12 data. The low absolute values of the
evaluation results do not necessarily imply that Sentinel-3B dataset is unreliable. Additionally, due to algorithmic reasons, grid
cells with outlier data were excluded. To highlight relevant information, Fig.8 only includes the results of Sentinel-3B data in
the interval distributions (e) and (f) while omitting the global distribution of the metrics (which mostly perform worse). It
should be noted that tCC and tRMSE mentioned in Section 3.4 are different from those in Section 3.3. The metrics in Section
3.4 can only reflect the relative ranking.

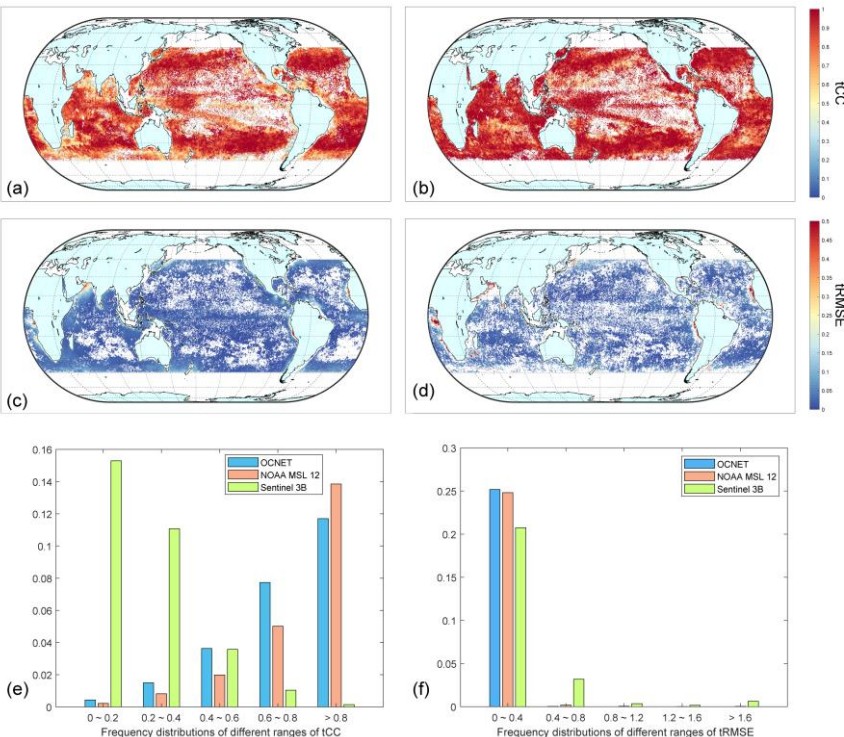

**Figure 8 Evaluation results based on the ETC evaluation method. Global distribution map of the tCC of Chl-a for a) OCNET and
b) NOAA MSL12. Global distribution map of the tRMSE of Chl-a for c) OCNET and d) NOAA MSL12. Interval distribution of e)
tCCs and f) tRMSEs for the three products was calculated using the ETC method.**

It can be seen that the output data of the OCNET model show a similar distribution to NOAA MSL12 data in the global tCC
distribution (Fig.8(a–b)), with most regions above 0.7. In the interval distribution (Fig.8e), the proportion of OCNET model



output data exceeding 0.8 is approximately 12%, slightly lower than NOAA MSL12's 14%. Regions with poorer tCC
       evaluation results are mainly distributed in the central Pacific Ocean where there are significant missing values in tCC, and
       the performance of OCNET is slightly lower than other regions. It should be noted that in the ocean areas near the American
       continent, there is a prevalent occurrence of tCC values below 0.5. This is similar to the evaluation results in Section 3.3, where
       the training performance for Region 9 is also slightly lower than other regions (Fig.7). Additionally, in the southern hemisphere
region of the Atlantic Ocean, OCNET seems to exhibit higher tCC values in the middle compared to the northern and southern
       sides, which is a different characteristic from the NOAA MSL12 dataset.

       From the results of tRMSE, the model output of OCNET is slightly better than NOAA MSL12 data. Specifically, NOAA
       MSL12 exhibits poorer tRMSE performance in the sea area near the west coast of Africa. This area is also characterized by
       high Chl-a concentration and significant interannual variations (Fig.3 and Fig.4). While OCNET exhibits similar high tRMSE
values in the ocean areas near the western side of Africa as NOAA MSL12, the distribution range is smaller compared to
       NOAA MSL12. Additionally, in the ocean areas near South America, both OCNET and NOAA MSL12 show small-scale high
       tRMSE values. It is worth mentioning that even for Sentinel-3B, the majority of tRMSE values are concentrated below 0.4.
       This may be related to the fact that most of the ocean Chl-a concentrations are relatively low.

## 4. Discussion

### 405    4.1 Factors affecting the distribution of marine phytoplankton

We focused on the surface chlorophyll-a (Chl-a) concentration as an indicator of the distribution of phytoplankton in the ocean
surface layer. The distribution of marine phytoplankton is influenced by various factors, including light, temperature, nutrients,
salinity, hydrodynamic conditions, and biological interactions (Behrenfeld et al., 2006; Ducklow et al., 2022; Feng et al., 2021).
Among them, light, temperature, salinity, and hydrodynamics are directly reflected in the input data of the OCNET model.
However, the influences of nutrients and biological interactions are more complex. Different phytoplankton communities
require different major nutrients such as nitrogen, phosphorus, and silicon (Powell et al., 2015; Takeda, 1998). The biological
interactions also include predation by zooplankton and the impact of human activities in coastal areas. Due to the lack of
publicly available reliable quantitative data on these two aspects, they are not considered in this study.

Considering the correlation between SST, SAL, and PAR with the growth cycle of phytoplankton, when creating input data
samples for the OCNET model, the mean values from one-month prior were selected as variables. However, hydrodynamic
conditions have real-time effects on the distribution of planktonic algae, so SSP and SST were taken as daily values for input.
It is worth mentioning that surface wind speed variations also have a direct impact on the movement of surface phytoplankton
in the ocean. However, wind speed not only includes direction and magnitude, but it also fluctuates significantly in both
direction and magnitude within a day. Therefore, simply taking daily averages as model inputs would not suffice. Additionally,
selecting too many variables can lead to overfitting or poor training performance due to limitations in the quantity of sample
data.



According to the result of OCNET model, it can be observed that regions with higher Chl-a concentration are generally located near continents (Fig.3). From the perspective of hydrodynamic conditions, hydrological factors such as water currents, ocean currents, and tides have a significant influence on the distribution and aggregation of phytoplankton. They affect the horizontal
migration, vertical mixing, and nutrient transport of phytoplankton. The nearshore waters have relatively low seawater velocity, coupled with features such as coastlines, underwater ridges, and archipelagos, which to some extent contribute to the retention and aggregation of phytoplankton. In addition, river inflows into the ocean often bring abundant nutrients (Slomp, 2011; Wang et al., 2016), creating favorable conditions for the growth of phytoplankton (Liu et al., 2022).

Global variations in ocean temperature also have an important impact on the growth of phytoplankton. With the continued
increase in global sea temperatures, temperature anomalies can also lead to anomalies in Chl-a concentration (Liu et al., 2022; Gruber et al., 2021; Le Grix et al., 2021). Global ocean warming results in more pronounced stratification of the ocean, altering the depth of the mixed layer and reducing vertical mixing between the surface layer and the cold, nutrient-rich layer below (Liu et al., 2022; Le Grix et al., 2021). The reduction in nutrients ultimately leads to a decrease in Chl-a concentration in the ocean surface layer. However, the declining trend in Chl-a concentration over the 20 years does not necessarily indicate a
reduction in algal blooms. On the contrary, the frequency of extreme events associated with algal blooms may be continuously increasing due to the influence of climate change (Feng et al., 2021; Dai et al., 2023).

In conclusion, understanding and studying these influencing factors are crucial for comprehending the ecological and biogeochemical processes of marine phytoplankton. A thorough investigation of the interactions among these factors can lead to better predictions and explanations of the growth and distribution patterns of phytoplankton. Subsequent research can further
focus on the impact of human activities in coastal areas on the growth of marine phytoplankton.

## 4.2 Uncertainty in ocean color data from satellite remote sensing

Satellite remote sensing is one of the important technologies to obtain long-term and large-scale ocean color data (Groom et al., 2019). However, there is uncertainty in the satellite data inversion process, degrading the accuracy (Groom et al., 2019; Hu et al., 2019a; Jiang and Wang, 2013). The first factor is the influence of atmospheric correction algorithms. The selection
of models and parameters can affect the satellite data during the atmospheric correction process. In addition, coastal areas and inland lakes closer to land often have more turbid waters, and the presence of high concentrations of suspended particles in complex water environments makes it more difficult for satellites to accurately retrieve water color information from the water surface (Lian et al., 2021; Wang et al., 2021; Zheng and Digiacomo, 2017). Furthermore, weather and environmental factors such as clouds and fog can partially or completely obscure the target water areas, posing important challenges to satellite data
acquisition (Zheng and Digiacomo, 2017; Wang et al., 2021).

In practical applications, in-situ measurements are typically used to calibrate and fit parameters for satellite data (Hu et al., 2012). However, obtaining a large amount of continuous shipborne measurement data is challenging, and publicly available in-situ data often suffer from problems such as inconsistent formats, varying measurement standards, complex composition of research institutions, and unclear data quality. Therefore, the application of in-situ data is limited for studying large-scale,



long-term time series of Chl-a concentration variations. In this study, to further demonstrate the data quality of OCNET outputs comparable to NOAA MSL12, an indirect evaluation method using ETC was employed. The evaluation results not only confirmed the excellent training performance of the OCNET model but also indicated significant differences among different satellite products, as evidenced by the low evaluation indicators for Sentinel-3B (Fig.7). Therefore, when applying satellite products of Chl-a concentration data, it is important to carefully select and correct for biases (Krug et al., 2017). It should be

noted that the global distribution map of ETC evaluation results shows a significant number of missing values (Fig.7). These missing values primarily stem from data gaps in Sentinel-3B and issues within the algorithm itself, resulting in negative squared evaluation metrics. Short data sequences or data that do not conform to the algorithm's underlying assumptions can lead to unusable results from the ETC algorithm. The final result analysis is based solely on grid cells with valid values.

The main purpose of our study was to address the serious issue of spatial missing values in existing satellite datasets. It should

be noted that the satellite-derived water color data itself still have errors that are difficult to correct (Wang et al., 2021). To improve accuracy, algorithms can be applied to differentiate different concentrations, such as OCx and CI algorithms (Hu et al., 2012), or specific parameter fitting can be performed for different regions (Li et al., 2019). However, the accuracy of satellite sensors, resolution, and other factors still influence the inversion accuracy. The accuracy of satellite data is not the focus of this study. Nevertheless, satellite data can still provide important references for algal blooms on a global scale (Wang

et al., 2021; Feng et al., 2021). An anomaly algorithm can also be used to reduce the impact of systematic biases (Wang et al., 2021; Stumpf, 2001). It may be beneficial to employ machine learning techniques for anomaly of Chl-a concentration, enabling better prediction of extreme events.

## 4.3 Applications of OCNET in the future

The variation of Chl-a concentration in the global ocean surface is influenced by various complex factors, which poses

challenges to accurately retrieve Chl-a concentration. We selected Chl-a data products retrieved from satellite data as a reference, supplemented by reanalysis data to provide environmental factor information. By combining the advantages of machine learning in big data analysis and simulation, we ultimately reconstructed a global-scale, long-term time series of Chl-a concentration dataset.

It is worth noting that this study intentionally excluded coastal regions in the selection of the study region, due mostly to the

poor performance of satellite data in coastal regions. Currently, most satellite data algorithms for Chl-a retrieval are based on the absorption peak of Chl-a in the blue spectral band (Hu et al., 2019b; Hu et al., 2012). This approach is highly applicable in open waters but can be significantly affected by interference in coastal regions, particularly in cases of high suspended matter concentration or colored dissolved organic matter (CDOM) (Blondeau-Patissier et al., 2014). Although adjustments can be made to the retrieval algorithms based on localized measurements, there is significant variability in water composition across

different coastal regions. This has resulted in poor performance of current satellite retrieval algorithms for estimating global Chl-a concentrations in coastal areas (Dai et al., 2023).



The performance of OCNET in coastal areas is primarily limited by the quality of the input satellite data. The construction of the OCNET model can be affected if the training-set quality is poor or severely lacking. However, OCNET has demonstrated its potential application in open waters. In regional calculations, OCNET can effectively capture the interrelationships among

various environmental factors in different zones and apply them to the reconstruction of Chl-a concentrations. There have also been successful studies applying machine learning to analyze Chl-a concentration variations at the regional scale (Chen et al., 2019; Roussillon et al., 2023), further demonstrating the potential application of machine learning methods in coastal areas. In the future, if reliable water color data from coastal areas can be obtained with a certain time span and spatiotemporal continuity for training OCNET, the reconstruction of Chl-a concentrations in coastal regions may also yield favorable results.

Overall, OCNET is capable of surpassing traditional machine learning methods such as multiple linear regression and random forest, as well as traditional artificial neural networks, because it can learn complex nonlinear relationships and incorporate global context into its predictions. This is of great significance for in-depth understanding and analysis of variable changes under the complex environmental influences in the context of big data.

## 5. Data availability

The reconstructed Chl-a data are archived and available at https://doi.org/10.5281/zenodo.8105194 (Hong et al., 2023).

## 6. Conclusion

We developed the OCNET model for the purpose of reconstructing global ocean Chlorophyll-a (Chl-a) concentration data. Chl-a is an important indicator of the health and productivity of marine ecosystems, and accurate measurements of Chl-a concentrations are essential for understanding the dynamics of these systems. The OCNET model is based on a convolutional

neural network and considers a variety of environmental variables that are known to influence the growth and distribution of ocean phytoplankton, which are the primary producers of Chl-a.

Our results show that the OCNET model performs very well in reconstructing Chl-a concentrations, accurately capturing the temporal variations of these features. This suggests that the model has strong potential for use in large-scale ocean color data reconstruction, and may even be able to predict Chl-a concentration trends in response to changes in the environment. However,

we did observe that the model's performance was somewhat weaker in the eastern Pacific region compared to other areas. This may be due to specific climate characteristics that have a significant impact on phytoplankton growth and distribution or low quality of satellite-based dataset in this region.

Overall, the OCNET model represents an important step forward in the use of machine learning techniques for predicting and reconstructing Chl-a concentrations. The model's strong performance in all regions of the globe suggests that it could be a

valuable tool for understanding and predicting the dynamics of marine ecosystems on a global scale. OCNET and other machine learning tools will help us better understanding and predicting the change of marine phytoplankton under climate





change. It is hoped that the results of this study will be of interest and relevance to a wide range of researchers, policymakers, and managers involved in the monitoring and management of aquatic ecosystems.

## Author contributions

LD and HZ designed the research. HZ, LD, LX, and WY developed the approaches and data sets. HZ, LD, LX, WY, ZJ, MA and MM contributed to the analysis of results and writing of the paper.

## Competing interests

The authors declare that they have no conflict of interest.

## Acknowledgments

This study was supported by the National Water and Energy Center, United Arab Emirates University, through the Asian University Alliance (AUA) program (Grants 31R281-AUA-NWEC-4-2020 and 12R023-AUA-NWEC-4-2020). Reviewers and editors' comments on improving this study and manuscript will be highly appreciated. The authors thank the researchers and their teams for providing all the datasets used in this study.

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
