# Peer review of "A global daily gap-filled chlorophyll-a dataset in open oceans during 2001–2021 from multisource information using convolutional neural networks"

_Earth System Science Data, 2023_

## Referee Comment (RC2)

This manuscript describes a large dataset from 2001-2021 for global daily gap filled chlorophyll-a. The authors developed a convolutional neural network called OCNET to reconstruct global chlorophyll-a concentration in open oceans. This dataset is very useful and important for the scientific community. The manuscript in general is well written, but would benefit from some minor clarifications, and adjustments before publication.

**General Comments:**

In line 155, the authors stated that they excluded regions from seas with surface salinities below 25. On the other hand, the minimum value of salinity shown in table 2 is 0. How? Please clarify this in the text.

According to ESSD, the DOI of the dataset and its in-text citation must be given in the abstract (https://www.earth-system-science-data.net/submission.html). Please add them.

Please Provide more details in section 5 (data availability section) about the dataset which is very relevant for a data description paper. For example, you can explain how are the data organized in different files (per year or ..), and how are the files named (chl_OCNET_.. followed by Date (day, month, year). What are the type of files (".asc", or ".csv".. etc) , separator (if any)… etc.

**Specific Comments:**

Line 15 in the abstract: missing data not "data missing"

Line 24 in the abstract: phytoplankton biomass not " phytoplankton mass"

Line 24- 25 in the abstract: The authors state that the "OCNET model achieves good performance in the reconstruction of global ocean Chl-a concentration data... etc.". We don't know how the model perform in polar regions or high latitudes (higher than 25) or coastal areas. It would be more precise to use the term "global open ocean". The sentence should then become as follows: "the OCNET model achieves good performance in the reconstruction of global **open** ocean Chl-a … etc."

Line 25 in the abstract: "captures temporal variations". It is recommended to write spatio-temporal.

Line 125: "Four" not "three" environmental variables

Line 259: add degree (°) to "0.25"

Please ensure consistency in the terminology used throughout the text and in the figures. For example, in line 333 and 341, the terms 'training set,' 'validation set,' and 'test set' are used, while in Figure 6, they are referred to as 'training set,' 'validating set,' and 'testing set.

Line 346-347: I recommend that you use "compared to" instead of "while". The statement would become: "Based on the results of the OCNET model, regions 2, 3, and 5 show larger decreasing magnitudes, **compared to** other regions, which **also** exhibit a decreasing trend."

Line 369-370: I recommend that you use "compared to" instead of "while". The statement would then be: "From the results of bias, the training set shows a clear tendency of underestimation (Fig. 7d), compared to the validation and testing sets, which exhibit a less pronounced underestimation."

Line 372: global open ocean Chl-a concentration instead of "global Chl-a concentration"

I recommend that you change all statements starting with: "it can be seen" or "it can be observed" and ending with (figure #) to: "By referring to figure #", **or** "Figure # indicates/shows …. Etc". Below are some examples:

> *Line 387-388: "It can be seen that the output data of the OCNET model show a similar distribution to NOAA MSL12 data in the global tCC distribution (Fig.8(a–b))"*

> *Line 332: "it can be seen that the model performs well (Fig.6)"*

Line 335-336, the authors stated that OCNET performed well but shows poor performance in individual regions, and stated that "region 9, being mostly near the American continent, is heavily influenced by human activities, and the satellite data quality in coastal areas is also poorer...". Then, in their conclusion (Line 510-512), the authors concluded that "the model's performance was somewhat weaker in the eastern Pacific region compared to other areas. This may be due to specific climate characteristics that have a significant impact on phytoplankton growth and distribution or low quality of satellite-based dataset in this region". Isn't region 9 supposed to be the eastern Pacific? If yes, then please state the same reasons in both statements, and provide more details or examples on such climate characteristics that are specific to the eastern Pacific.

**Figure 6** shows the evaluation indices of the training, testing and validating sets. Meanwhile, there is no indication to which dataset correspond the evaluation metrics shown in **table 4**. Although can be inferred by comparison, I recommend that you indicate in the text or table caption that they correspond to the training set. Readers shouldn't guess.

Several References lack DOI. Please add the corresponding DOI. Below are **some** examples of references lacking DOI:

Behrenfeld, M. J., O'Malley, R. T., Siegel, D. A., McClain, C. R., Sarmiento, J. L., Feldman, G. C., Milligan, A. J., Falkowski, P. G., Letelier, R. M., and Boss, E. S.: Climate-driven trends in contemporary ocean productivity, Nature, 444, 752-755, 2006.

Chen, S., Hu, C., Barnes, B. B., Xie, Y., Lin, G., and Qiu, Z.: Improving ocean color data coverage through machine learning, Remote Sensing of Environment, 222, 286-302, 2019.

Groom, S., Sathyendranath, S., Ban, Y., Bernard, S., Brewin, R., Brotas, V., Brockmann, C., Chauhan, P., Choi, J.-k., Chuprin, A., Ciavatta, S., Cipollini, P., Donlon, C., Franz, B., He, X., Hirata, T., Jackson, T., Kampel, M., Krasemann, H., Lavender, S., Pardo-Martinez, S., Mélin, F., Platt, T., Santoleri, R., Skakala, J., Schaeffer, B., Smith, M., Steinmetz, F., Valente, A., and Wang, M.: Satellite Ocean Colour: Current Status and Future Perspective, Frontiers in Marine Science, 6, 485 2019.

---

## Author Response (AR1)

Comment:

The variation of Chl-a concentration in the global ocean surface is influenced by various complex factors, which poses challenges to accurately retrieve Chl-a concentration. In this manuscript, the authors selected Chl-a data products retrieved from satellite data as a reference, supplemented by reanalysis data to provide environmental factor information. By combining the advantages of machine learning in big data analysis and simulation, they ultimately reconstructed a global-scale, long-term time series of Chl-a concentration dataset. The results show that the OCNET model performs very well in reconstructing Chl-a concentrations, accurately capturing the temporal variations of these features. This suggests that the model has strong potential for use in large-scale ocean color data reconstruction, and may even be able to predict Chl-a concentration trends in response to changes in the environment.

The dataset is extremely valuable for a wide range of researchers, policymakers, and managers involved in the monitoring and management of aquatic ecosystems. This study and its results are really novel and impressive to me.

Overall, I find this study and its results to be highly promising and valuable for the scientific community. With the suggested clarifications and improvements, this manuscript has the potential to make a significant contribution to the field of Chl-a concentration retrieval and ocean color data reconstruction.

Response:

We really appreciate these overall comments and recommendation by this reviewer. Our point-by-point responses to the reviewer's comments are given as follows.

Specific Comments:

1) Pg. 3, Lines 65-66: Please provide an explanation for the difference between OCNET and the CNNs used in previous studies (Cao et al., 2020; Jin et al., 2021; Cen et al., 2022; Yussof et al., 2021). Clarifying this distinction would enhance the reader's understanding of the novelty of the OCNET model.

Response:

Thanks for this comment. OCNET, a modified U-Net architecture, deviates from the general Convolutional Neural Network (CNN) in terms of network structure. Most data reconstruction methods based on machine learning, including CNNs and random forest, primarily rely on the spatiotemporal correlations within the data, using valuable spatiotemporal sequences to infer missing regions. However, such methods struggle to achieve satisfactory results when dealing with extensive, irregularly distributed missing data. The OCNET machine learning method proposed in this study draws inspiration from oceanic biogeochemical models and incorporates environmental variables that influence the distribution of chlorophyll-a concentration into data reconstruction. By learning the impact of environmental factors on the variation of chlorophyll-a concentration, it enables long-term and large-scale reconstruction of missing data. We will further elaborate on the distinctive features of OCNET in the revised manuscript.

Modifications: The difference between OCNET and other CNNs mentioned in previous studies were

added in lines 65-75.

2) Pg. 6, Lines 125-126: The authors state that "We have selected three environmental variables, i.e., sea surface temperature (SST), salinity (SAL), and photosynthetically active radiation (PAR) as the input data for the OCNET model." However, it appears that the input data also contain SSP. Please address this discrepancy and provide clarity on whether SSP is included in the input data or not.

Response:

Thanks for this comment. In fact, we did select four environmental variables, including SSP, as inputs. The three variables mentioned here (SST, SAL, and PAR) are directly related to phytoplankton growth, as further explained in the manuscript. SSP primarily reflects the dynamic environmental factors at the ocean's surface, which influence the spatial distribution of phytoplankton. To eliminate any potential ambiguity, we have made corresponding modifications in the manuscript at the relevant sections.

Modifications: The description of the input data was clarified in lines 133-141.

**Response to community comment 1**

Comment:

I agree with reviewer #1 that this dataset has the potential to be extremely valuable (I found it while searching for a gapless daily chl dataset to test an idea on), and I thank the authors for making their data publicly available. However, I'm leaving this comment to suggest that the authors provide the data as netCDF files, perhaps one for each year, which would meet this journal's request that the data be provided in a non-proprietary community-established format that is findable, accessible, interoperable, and reusable. In the repository linked in the manuscript, the data are provided as 74 .rar files, each containing a number of ascii files. Every interested user (including myself, right now) will have to download and extract all 74 rar files, then write their own code to read the non-standard format and take a guess at some of the missing metadata (e.g., units).

Response:

Thank you very much for your suggestions. It is important for us to receive these feedbacks to further improve the data set. The new version of the dataset has been re-uploaded. Please refer to https://doi.org/10.5281/zenodo.10011908. This version of the dataset consists of one NetCDF file per year. Information regarding data format, latitude and longitude, handling of missing values, units, and other data specifications has been incorporated into the netCDF files. Subsequent updates to the dataset DOI will be provided in the revised manuscript. Thank you once again for your valuable suggestions!

Modifications: DOI of new version of OCNET dataset was updated in abstract and section 5.

**Response to referee comment 2**

Comment:

This manuscript describes a large dataset from 2001-2021 for global daily gap filled chlorophyll-a. The authors developed a convolutional neural network called OCNET to reconstruct global chlorophyll-a concentration in open oceans. This dataset is very useful and important for the scientific community. The

manuscript in general is well written, but would benefit from some minor clarifications, and adjustments before publication.

Response: Thanks for thoroughly reviewing the manuscript and making such encouraging comments. It is important for us to receive these feedbacks to further improve the data set and the manuscript. Comments and issues mentioned in referee comment 2 have been addressed and are illustrated as follows.

General Comments:

In line 155, the authors stated that they excluded regions from seas with surface salinities below 25. On the other hand, the minimum value of salinity shown in table 2 is 0. How? Please clarify this in the text. According to ESSD, the DOI of the dataset and its in-text citation must be given in the abstract (https://www.earth-system-science-data.net/submission.html). Please add them. Please Provide more details in section 5 (data availability section) about the dataset which is very relevant for a data description paper. For example, you can explain how are the data organized in different files (per year or ..), and how are the files named (chl_OCNET_.. followed by Date (day, month, year). What are the type of files ("asc", or "csv".. etc) , separator (if any)… etc

Response:

Thanks for this comment. When defining the open ocean areas, we used the multi-year mean of WOA2013 data as the reference and selected regions where salinity is greater than 25 PSU. However, it is important to note that the seasonal and interannual variations in salinity in these regions may not always exceed 25 PSU. It should be emphasized that, to establish the study boundaries, it is not a strict requirement for salinity to always exceed 25 PSU, but rather for the mean value to meet this criterion.

Furthermore, the study utilized salinity data from the Ocean Reanalysis System 5, and according to the documentation for this dataset, the minimum salinity value is 0. Therefore, in Table 2, the salinity is described as having a minimum value of 0.

Thanks for the comment about the dataset information. The DOI for the dataset will be added in the abstract, and any updates to the dataset will be provided through the same link. Please refer to the instructions on the dataset's publishing website for details (the data sets are available online with a DOI: https://doi.org/10.5281/zenodo.10011908).

Initially, the first version of the dataset was released in ASC format with daily data files. Due to the high volume of files in the first version, we have opted to release a new version and have converted the data into netCDF files. In the second version, each file corresponds to one year of data, and the respective year is indicated in the file name. Information regarding data format, latitude and longitude, handling of missing values, units, and other data specifications has been incorporated into the netCDF files.

Modifications: The source of the minimum value is added in title of Table 2. DOI of new version of OCNET dataset was updated in abstract and section 5.

Specific Comments:

1) Line 15 in the abstract: missing data not "data missing"

Response:

Thanks for this comment. It will be corrected in the revised manuscript.

2) Line 24 in the abstract: phytoplankton biomass not "phytoplankton mass"

Response:

Thanks for this comment. It will be corrected in the revised manuscript.

3) Line 24-25 in the abstract: The authors state that the "OCNET model achieves good performance in the reconstruction of global ocean Chl-a concentration data... etc.". We don't know how the model perform in polar regions or high latitudes (higher than 25) or coastal areas. It would be more precise to use the term "global open ocean". The sentence should then become as follows: "the OCNET model achieves good performance in the reconstruction of global open ocean Chl-a … etc."

Response:

Thanks for this comment. It will be corrected in the revised manuscript.

4) Line 25 in the abstract: "captures temporal variations". It is recommended to write spatiotemporal.

Response:

Thanks for this comment. It will be corrected in the revised manuscript.

5) Line 125: "Four" not "three" environmental variables

Response:

Thanks for this comment. There is indeed an issue with clarity in the text. What we intended to convey is that the study selected three variables that affect the growth of marine phytoplankton, namely SST, SAL, and PAR, along with one variable that influences their distribution, namely SSP. SST, SAL, and SSP are derived from reanalysis data, while PAR is obtained from satellite data products. To address this, we have made appropriate revisions to the data description section from lines 125 to 135 to clarify this point.

Modifications: The description of the input data was clarified in lines 133-141.

6) Line 259: add degree (°) to "0.25"

Response:

Thanks for this comment. It will be corrected in the revised manuscript.

7) Please ensure consistency in the terminology used throughout the text and in the figures. For example, in line 333 and 341, the terms 'training set,' 'validation set,' and 'test set' are used, while in Figure 6, they are referred to as 'training set,' 'validating set,' and 'testing set.'

Response:

Thanks for this comment. It will be corrected in the revised manuscript.

Modifications: The terminology used for the training set, validating set, and testing set has been standardized throughout the entire manuscript.

8) Line 346-347: I recommend that you use "compared to" instead of "while". The statement would

become: "Based on the results of the OCNET model, regions 2, 3, and 5 show larger decreasing magnitudes, compared to other regions, which also exhibit a decreasing trend."

Response:

Thanks for this comment. It will be corrected in the revised manuscript.

9) Line 369-370: I recommend that you use "compared to" instead of "while". The statement would then be: "From the results of bias, the training set shows a clear tendency of underestimation (Fig. 7d), compared to the validation and testing sets, which exhibit a less pronounced underestimation.

Response:

Thanks for this comment. It will be corrected in the revised manuscript.

10) Line 372: global open ocean Chl-a concentration instead of "global Chl-a concentration"

Response:

Thanks for this comment. It will be corrected in the revised manuscript.

11) I recommend that you change all statements starting with: "it can be seen" or "it can be observed" and ending with (figure #) to: "By referring to figure #", or "Figure # indicates/shows …. Etc". Below are some examples:

> Line 387-388: "It can be seen that the output data of the OCNET model show a similar distribution to NOAA MSL12 data in the global tCC distribution (Fig.8(a–b))"

> Line 332: "it can be seen that the model performs well (Fig.6)

Response:

Thanks for this comment. It will be corrected in the revised manuscript.

Modifications: The statements ending with (figure #) were changed into "referring to figure #", or "Figure # indicates/shows …. Etc" everywhere.

12) Line 335-336, the authors stated that OCNET performed well but shows poor performance in individual regions, and stated that "region 9, being mostly near the American continent, is heavily influenced by human activities, and the satellite data quality in coastal areas is also poorer...". Then, in their conclusion (Line 510-512), the authors concluded that "the model's performance was somewhat weaker in the eastern Pacific region compared to other areas. This may be due to specific climate characteristics that have a significant impact on phytoplankton growth and distribution or low quality of satellite-based dataset in this region". Isn't region 9 supposed to be the eastern Pacific? If yes, then please state the same reasons in both statements, and provide more details or examples on such climate characteristics that are specific to the eastern Pacific.

Response:

Thanks for this comment. In these two sentences, there is indeed a lack of clarity in our statements. In fact, the summaries of these two sections refer to slightly different regions. Region 9 encompasses parts of the Eastern Pacific and the western North Atlantic, all of which are close to the American continent.

However, when we mention that "the model performs poorly in the eastern tropical Pacific," it actually refers to the "eastern tropical Pacific," which is closer to the equator (as shown in Figure 7) and should be clarified in the revised manuscript.

Since the OCNET model proposed in the study does not consider the influence of human activities in coastal areas, the poor performance of the model in the coastal regions near the American continent could likely be attributed to human activities. On the other hand, the model's poor performance in the "eastern tropical Pacific" region may be more likely to be affected by specific climate characteristics. These unique climate variations might not be captured by the OCNET model within the relatively short training time span (2018-2021). In fact, the results of the OCNET model for Region 9 during the period of 2018-2021 closely align with the satellite-merged OCCCI dataset. This suggests that OCNET performs well only within the time frame covered by the target dataset NOAA MSL12, and it exhibits poorer performance in earlier periods (2001-2017).

There are several studies that focus on anomalies in the eastern tropical Pacific. For example, Geng et al. suggest that increased sea surface temperature variability due to global warming may manifest in the eastern Pacific earlier than central Pacific. Duteil et al. discuss the important impact of future changes in atmospheric synoptic variability (ASV) on ocean properties and primary productivity in the eastern tropical Pacific.

Regarding the issue of the model's poor performance in this particular region, we plan to conduct further analysis and exploration in future work. Once again, we appreciate your feedback!

Modifications: In the conclusion section, "eastern Pacific" was changed into "eastern tropical Pacific". The citations for the two mentioned articles have also been included in the manuscript.

13) Figure 6 shows the evaluation indices of the training, testing and validating sets. Meanwhile, there is no indication to which dataset correspond the evaluation metrics shown in table 4. Although can be inferred by comparison, I recommend that you indicate in the text or table caption that they correspond to the training set. Readers shouldn't guess.

Response:

Thanks for the comment. We indeed omitted an explanation of the datasets included in the evaluation metrics in Table 4. The median values in these metrics represent the median of all evaluation results, including those from the training set, validation set, and test set. Since Figure 6 already provides separate visualizations of the evaluation results for the training set, validation set, and test set using box plots, Table 4 only shows the overall evaluation summary. We have added an explanation to the title of Table 4 to clarify this.

Modifications: The explanation of evaluation metrics in Table 4 was added in lines 339-340.

14) Several References lack DOI. Please add the corresponding DOI. Below are some examples of references lacking DOI:

Behrenfeld, M. J., O'Malley, R. T., Siegel, D. A., McClain, C. R., Sarmiento, J. L., Feldman, G. C., Milligan, A. J., Falkowski, P. G., Letelier, R. M., and Boss, E. S.: Climate-driven trends in contemporary ocean productivity, Nature, 444, 752-755, 2006. Chen, S., Hu, C., Barnes, B. B., Xie, Y., Lin, G., and Qiu,

Z.: Improving ocean color data coverage through machine learning, Remote Sensing of Environment, 222, 286-302, 2019. Groom, S., Sathyendranath, S., Ban, Y., Bernard, S., Brewin, R., Brotas, V., Brockmann, C., Chauhan, P., Choi, J.-k., Chuprin, A., Ciavatta, S., Cipollini, P., Donlon, C., Franz, B., He, X., Hirata, T., Jackson, T., Kampel, M., Krasemann, H., Lavender, S., Pardo-Martinez, S., Mélin, F., Platt, T., Santoleri, R., Skakala, J., Schaeffer, B., Smith, M., Steinmetz, F., Valente, A., and Wang, M.: Satellite Ocean Colour: Current Status and Future Perspective, Frontiers in Marine Science, 6, 485 2019.

Response:

Thanks for the comment. The DOI of all references will be modified in the revised manuscript.

A list of modifications in the manuscript

Modification position is referred to the marked-up manuscript

| Comment | Modification | Modification position |
|---|---|---|
| Modifications based on RC1 | | |
| RC1 General comment | None | None |
| RC1 Specific(1) | The difference between OCNET and other CNNs mentioned in previous studies were added in section 1 | P3, L68-L74 |
| RC1 Specific(2) | The description of the input data was clarified in section 2.1 | P7, L134-L142 |
| Modifications based on CC1 | | |
| CC1 General comment | The new version of the dataset has been re-uploaded as netCDF files. Please refer to https://doi.org/10.5281/zenodo.10011908. | None |
| Modifications based on RC2 | | |
| RC2 General comment1 | None | None |
| RC2 General comment2 | The source of the minimum value is added in title of Table 2. The new version of the dataset has been re-uploaded as netCDF files and the DOI is updated in the revised manuscript. (https://doi.org/10.5281/zenodo.10011908) | P1, L27 P8, L189-L190 P25, L525 P27, L618-L619 |
| RC2 Specific(1) | "data missing" was changed into "missing data" | P1, L15 |
| RC2 Specific(2) | "phytoplankton mass" was changed into "phytoplankton biomass" | P1, L25 |

| | | |
|---|---|---|
| RC2 Specific(3) | "global ocean" was changed into "global open ocean" | P1, L25 |
| RC2 Specific(4) | "tenporal" was changed into "spatiotemporal" | P1, L26 |
| RC2 Specific(5) | The description of the input data was clarified in section 2.1 | P7, L134-L142 |
| RC2 Specific(6) | The unit degree ( ͦ) was added | P12, L271 |
| RC2 Specific(7) | The terminology used for the training set, validating set, and testing set has been standardized throughout the entire manuscript. | Everywhere |
| RC2 Specific(8) | "while" was changed into "compared to" | P18, L364 |
| RC2 Specific(9) | "while" was changed into "compared to" | P20, L389 |
| RC2 Specific(10) | "global Chl-a" was changed into "global open ocean Chl-a" | P20, L391-L392 |
| RC2 Specific(11) | The statements ending with (figure #) were changed into "referring to figure #", or "Figure # indicates/shows …. Etc" everywhere. | Everywhere |
| RC2 Specific(12) | In the results section and conclusion section, "eastern Pacific" was changed into "eastern tropical Pacific". The citations for the two mentioned articles have also been included in the manuscript. | P20, L381 P21, L411-L412 P25, L535-L538 |
| RC2 Specific(13) | The explanation of evaluation metrics in Table 4 was added. | P17, L346-L347 |
| RC2 Specific(14) | The DOI of all references was added in the revised manuscript. | References |

[revised manuscript text omitted]